# Status of the stateless population in Thailand: How does stigma matter in their life?

**Siwarak Kitchanapaibul**[1,2]**, Tawatchai Apidechkul**[1,2]*****, Peeradone Srichan**[1]**, Thanatchaporn Mulikaburt**[1]**, Onnalin Singkhorn**[2,3]**, Anusorn Udplong**[1]**, Panupong Upala**[2]**, Chalitar Chomchoei**[2]**, Fartima Yeemard**[2]**, Ratipark Tamornpark**[1,2]**, Pilasinee Wongnuch**[1,2]

**1** School of Health Sciences, Mae Fah Luang University, Chiang Rai, Thailand, **2** Center of Excellence for Hill Tribe Health Research, Mae Fah Luang University, Chiang Rai, Thailand, **3** School of Nursing, Mae Fah Luang University, Chiang Rai, Thailand

* Tawatchai.api@mfu.ac.th

## Abstract

### Background

The stateless population in Thailand live by accessing all public services, including the health care system. Stigma is a crucial factor impacting these individuals' lives and their access to medical care. This study aimed to understand the experience of the stateless population in Thailand and how they overcome the problem of stigma.

### Methods

A qualitative method was used to elicit information from key informants who were members of the stateless population, which was classified as those who did not hold Thai identification cards (IDs). A questionnaire was used to guide the interview, which was conducted in a private and confidential room. The interviews were conducted after voluntary agreement was obtained from the participants; each interview was held in August 2021 and lasted for approximately 45 minutes.

### Results

Fifty-one people participated in the study; 68.6% were females, 86.3% were married, and 90.2% were Akha or Lahu. The stateless population in Thailand reported four types of perceived stigma: having a lower ability to request that their needs be met, not being equal to others, not being able to qualify for health care services and being ranked below other hill tribe people who have IDs. The phrase "life is nothing" was presented by the participants, who reported that they felt like an invisible population in Thailand. Some participants reported that other hill tribe people who have IDs act as stigma perpetuators among members of the stateless population who do not have IDs. Maintaining their privacy within their village, trying to obtain a Thai ID, and practicing the Thai language were the main ways of avoiding the stigma reported by the stateless population. Obtaining a Thai ID was detected as the top goal in their aim to overcome the stigma problem.

**Data Availability Statement:** All relevant data are within the paper and its Supporting Information files.

**Funding:** The study has been supported small grant from the center of excellence for the hill tribe

health research, Mae Fah Luang University with the number 2-2021. However, the funders had no role in study design, data collection and analysis, decision to publish, or preparation of the manuscript.

**Competing interests:** The authors have declared that no competing interests exist.

## Conclusions

The stateless population in Thailand live as an invisible population and are negatively treated via various patterns from others. Accessing Thai IDs and education are argued to be the most effective procedures for addressing the problem under the implementation schemes of the relevant organizations.

## Introduction

Stigma is defined as a negative social attitude attached to a characteristic of an individual who may be regarded as having a mental, physical, or social deficiency [1]. The impacts of stigma are present at different levels: an individual [2], a family [3] or a group of people who have specific characteristics [4, 5]. The impacts also involve several dimensions, such as an increasing lack of access to health care services [6, 7], facing a large barrier to job access [8], and having one's value as a human being minimized [9]. Stigma has been detected in all societies [10], but the impacts may be different. Some individuals must manage only one form of stigma [11], while others see stigma as having a massive impact on their lives [12]. While some stigma is present at the individual or family level, specific groups are faced with stigma as a whole merely by virtue of belonging to the group, such as the stateless population in Thailand. In 2020, 479,943 people were classified as stateless people in Thailand [13]; these individuals live in distant and remote areas of Thailand where accessing information from the Thai government is very difficult [14].

Stigma refers to an attribute that is deeply discrediting or that enables varieties of discrimination that ultimately deny the individual or group full social acceptance and that reduce the individual's opportunity [15]. Prejudice is a result of the cognitive and affective responses to stereotypes [16]. Amiot and Bourhis [17] defined discrimination as positive or negative behaviors toward a social group and its members. Generally, the behavior is negative and can be classified as direct or indirect. Abrams [18] added that prejudice includes behaviors and opinions, while discrimination constitutes behaviors toward another group or person. Therefore, in this study, we aimed to understand the stigma that deeply interferes with the stateless population in Thailand.

To understand the context of the stigma facing the stateless population in Thailand, the framework integrating normative influence on stigma (FINIS) was applied [19]. Many theories and concepts attempt to explain stigma in various contexts. However, the FINIS is best able to explain the stigma experience in daily life among the stateless population. The FINIS focuses on three layers: the micro, meso, and macro. The microlayer focuses on individual characteristics, while the mesolayer focuses on family and community contexts. Finally, the macrolayer focuses on the stigma embedded in a larger national cultural context.

The stateless population in Thailand consist of a group of people who have migrated from neighboring countries to settle in Thailand, especially along the border areas of Thailand and Myanmar; they do not have Thai identification cards (IDs) [20]. Essentially, the stateless population in Thailand are often members of typical hill tribes, such as the Akha, Lahu, Hmong, Lisu, Yao, etc. [21]. These tribes have their own languages, clothing, and lifestyles, which specifically act as influencers of stigma from others [22]. Most of the stateless population in Thailand live in a poor socioeconomic situation [23] and have less opportunity than Thai citizens do to access schools [24]. This situation is largely because access to schools and employment at good professional jobs in Thailand require a Thai ID.

People living in Thailand who are not granted a Thai ID are identified as members of the stateless population [25]. The Thai ID is used to access all public services in Thailand and is officially used to identify those with Thai citizenship [26]. The card consists of 13 digits and is required to access all government procedures in Thailand. Moreover, it is used for access to education and to apply for a professional job, including purchasing real estate. The most important advantage of having a Thai ID is access to health care services. Therefore, the absence of Thai IDs among the stateless population leads to an inability to access almost all of the activities that support a good quality of life. Furthermore, those who lack Thai IDs become victims of stigma from people around them through several processes. The severity of such stigma minimizes the value of an individual's status as a human being.

The current study aimed to understand the stigma-related experiences of the stateless population living in Thailand and their approaches to managing the problem. The findings could be used to determine specific and powerful national and international policies aimed at improving the relevant systems, with the ultimate goal of supporting an improvement in the quality of life of these individuals.

## Methods

Qualitative and face-to-face in-depth interviews were used to gather information from the participants. A purposive selection was made of the participants who lived in the five remote hill tribe villages located along the border of Thailand and Myanmar. The participants were considered to meet the criteria if they were stateless people living in Thailand who did not hold a Thai ID and who had experienced any forms of stigma.

A questionnaire was developed during the literature review and subsequently discussed with the hill tribe village leaders, including six stateless people. The questions were assessed for appropriateness, including validity, by the research team and two experts working in the field (one medical anthropologist and one sociologist). The questionnaire consisted of six questions: 1) "What do you think about being a member of a 'stateless population'?" 2) "Have you experienced any events that have made you feel uncomfortable when you have contacted people or health care providers?" 3) "Could you please tell me about stigma-related situations that you face in your daily life?" 4) "How do these situations impact you and your life?" 5) "How do you cope with these situations?" 6) "What other expectations do you have about such a stigma?"

All the project concepts and tools were approved by the Chiang Rai Public Health Office Ethics Committee on Human Research (Committee; CRPPH0 No. 69/2564) before the project commenced. The five targeted hill tribe village leaders were contacted, and the contents of the study were explained to them, including the objectives. After being granted access to the villages, five days ahead of the desired date, an appointment was made to enter the village, and the village leaders were asked to introduce those people who did not possess Thai IDs to participate in the project. On the date of the data collection, all participants were confirmed to conform to the criteria of the study before engaging in the interview. Before the interviews, all participants were asked to provide informed consent. For those who could not speak Thai, public health volunteers from the village were asked to help translate all of the study contents before the participants fingerprinted the consent form. Afterward, the interviews were conducted in a private room at the village hall, which had previously been prepared by the village leaders.

Each interview began with an introduction of the interviewer, who had long-term experience working with qualitative methods and was familiar with this population after having worked with them for more than ten years. All interviews were recorded after obtaining approval from the participants. The interviews lasted approximately 45 minutes each.

All records were typed and checked for errors. The transcripts were determined to have reached the level of saturation by the research team. Each transcript was sent back to the related participant to confirm the accuracy of the information before further analysis. The information was coded and developed as a preliminary coding tree. All information was transferred into the NVivo program (NVivo, qualitative data analysis software; QSR International Pty Ltd., version 11, 2015) to derive the themes. The final themes were reconsidered among the research team before any conclusions were made. The findings were sent to the participants again before interpretations were drawn.

### Ethics approval and consent to participate

All research protocols and tools were reviewed and approved by the Committee (No. REH-70/2564). Before the interview, the essential information was clearly communicated in Thai and Burmese to the participants. Informed consent was obtained on a voluntary basis. Those who could write were asked to provide a fingerprint representing informed consent on a voluntary basis.

## Results

### I. General characteristics

Fifty-one participants were interviewed to extract information regarding the purpose of the study: 68.6% were females, 86.3% were married, and 90.2% were Akha or Lahu. The majority were Christian (72.6%), had no education (66.7%), and were farmers (56.9%), with a median monthly income of 2,888 baht on average ($93) (Table 1).

### II. Perceptions of stigma

There were different forms and levels of the participants' perceptions of stigma among the stateless population in Thailand: having a lower ability to request that their needs be met, not being equal to others, not being able to qualify for health care services and even being ranked below other hill tribe people who have IDs.

**a) Having a lower ability to request that their needs be met.** The stateless population felt that they did not have the power to negotiate and had no voice during everyday situations or when requesting needs, especially regarding access to health care services. This feeling was detected among almost all participants. Some people reported that they voiced their needs but acquired only a minimal response from health care providers. However, many people reported remaining silent about their needs because they feared that they would not receive good care from health care providers.

A 54-year-old woman stated the following [P#20]:

"I had a stomachache a month ago, and I went to see a doctor in a hospital. He prescribed some medicines to me, but they made me feel only a little better. I truly did not know what I should do next. I hoped I would be referred to a larger hospital in the city. However, I had no idea because I have no Thai ID. I do not have any right to ask for help from him. It is like no one can hear my voice."

A 53-year-old woman stated the following [P#11]:

"A few months ago, I got a bad headache; my expectation was to get good treatment from a big hospital in Chiang Rai, but I did not dare to tell the staff about my expectation. I was afraid that if I said something wrong, I would not get good care from them. I kept it to myself even though I truly wanted to get care from a better hospital."

A 30-year-old woman stated the following [P#15]:

**Table 1. Demographic data of the participants.**

| Characteristics | n | % |
|---|---|---|
| **Total** | 51 | 100.0 |
| **Sex** | | |
| Male | 16 | 31.4 |
| Female | 35 | 68.6 |
| **Age** Min = 15, Max = 70, Mean = 43 years | | |
| **Marital status** | | |
| Single | 3 | 5.9 |
| Married | 44 | 86.3 |
| Previously married | 4 | 7.8 |
| **Tribe** | | |
| Thai Yai | 5 | 9.8 |
| Lahu | 20 | 39.2 |
| Akha | 26 | 51.0 |
| **Religion** | | |
| Buddhist | 14 | 27.4 |
| Christian | 37 | 72.6 |
| **Education** | | |
| No education | 34 | 66.7 |
| Primary school | 12 | 23.5 |
| Secondary school | 4 | 7.8 |
| Diploma school | 1 | 2.0 |
| **Occupation** | | |
| Farmer | 29 | 56.9 |
| Daily wage employee | 14 | 27.4 |
| Unemployed | 7 | 13.7 |
| In school | 1 | 2.0 |
| **Income** (baht) Min = 0, Max = 9,500, Median = 2,888 baht | | |

"When I was at the hospital for the delivery of my baby 3 years ago, I screamed very loud because of the pain, and a nurse came and shouted at me. I stopped and kept quiet even though the pain worsened. I could not ask for any help from her. I was afraid that I would be treated badly."

Many stateless people have previously been or currently are faced with a limited ability to state their needs and request that they be met. This experience was reported by almost all of the stateless people we interviewed.

**b) Being unequal to others.** This form of stigma was reported in particular by young adult members of the stateless population. The sense was that stateless individuals are not treated as though they are equal to other people, especially by other hill tribe people who hold Thai IDs and are thus confirmed to be Thai citizens. The perception of not being equal to others was also reported in relation to attending health care services under different scenarios, such as delays in seeing a doctor and repeatedly being queued.

A 15-year-old woman stated the following [P#1]:

"I think nothing is equal in my life. This has always happened to me and my family. I have experienced it since I was young. One day, my grandma got ill, and I brought her to a hospital. I informed a nurse and waited for a long time before getting help from her. Neither of us could speak Thai, so my grandma was queued behind everyone else and got worse. I think about my bad experience a lot. It hurts me so much."

A 20-year-old woman stated the following [P#25]:

"My main illness is migraines. I have to see my doctor monthly, and I am always the last person being cared for. I accept that because I have no Thai ID and so things are not equal for me.

These population members have faced unequal situations when trying to access health care services. Most of them have experienced delays in seeing a doctor and have many unanswered questions in their minds, but only non-Thai citizenship is given as a reason for such inequality.

**c) Not being able to qualify for health care services.** All members of the stateless population are charged when they go to any public or private hospital in Thailand, except for some basic public health programs, such as immunizations for children, because they do not have Thai IDs. The card is used to gain free access to all public services and to be considered as having full rights under the Thai Constitution of the Kingdom of Thailand. Being charged medical fees before being able to obtain such services was one of the bad experiences reported among the stateless population. Experiences with being confronted by some typical words or sentences, including physical language from health care providers, were one of the issues that caused the most suffering, according to the stateless population. "A man who was not fully a man" was the definition of their situation because they did not have the full qualifications to access health care services.

A 63-year-old man stated the following [P#13]:

"Without an ID, I feel that I am not fully a man. I am so jealous of those who have an ID. If you have a Thai ID, you have lots of advantages. I feel like I cannot get even essential things in the same way as others. I have now been diagnosed with hypertension. Do you know how much I suffer to see a doctor to get a prescription every month? I have to pay so much money for my medicine. I suffer a lot."

A 44-year-old woman stated the following [P#17]:

"I had a very bad experience with this hospital. I have money to pay my medical fees, but they still acted like I was a very poor just because I do not have an ID. You know a nurse said to me, "Do you have money to pay for these services? It is expensive for you, you know?" I think because I have no Thai ID, I have been labeled as poor."

A 47-year-old woman stated the following [P#21]:

"I have had diabetes mellitus for 2 years. The nurse told me that because of my disease, I need to take pills for the rest of my life. I think if I could go to a better hospital, I would be cured, but the problem is that I do not have a Thai ID. I also do not think I am qualified for access to a big hospital in the city. I also fear that because I am not Thai, they might not take good care of me."

One of the main barriers to health care services is the lack of Thai IDs. Those members of the stateless population who do not have a Thai ID do not qualify for access to free medical and health care services although every country must treat all humans equally.

**d. Being ranked below other hill tribe people who have IDs.** A large proportion of the hill tribes in Thailand have obtained Thai IDs, which means that they are considered full Thai citizens. Individuals with Thai IDs are members of families who immigrated into Thailand centuries ago and are now in their third or fourth generation. In contrast, the stateless population members only recently moved to Thailand. Many stateless people feel that they live at a standard below that of the tribe people who have access to health care services. Along with successful health care services, other services are granted at a lower priority to tribe people who do not hold Thai IDs. The quotes below provide more information about being provided with services that are secondary in nature compared to those who are in the same tribe but hold a Thai ID.

A 63-year-old man stated the following [P#13]:

"I and some other tribes came from the same original area in Myanmar many years ago. In my mind, we should be the same; but you know, we are not the same. I have no Thai ID, but they do have one. They have full rights as Thai citizens, but I do not. I feel I am standing below everyone . . . I am lower than some hill tribe people who have IDs. The card makes us different."

A 30-year-old woman stated the following [P#15]:

"Two years ago, I went to a hospital for antenatal care. While I was waiting in the public waiting room, a hill tribe woman presented her Thai ID to the hospital staff. She was very proud because she did not have to pay her hospital fees. I feel that I am lower than everyone. I am different from other tribes who have IDs; I am lower than them."

Hierarchies of power occur when social values cause differences to exist between individuals. The stateless population have realized that they are positioned lower than the hill tribe members who have been granted Thai citizenship. This situation means that obtaining IDs could increase these individuals' power.

**e. No better options in life.** As members of a stateless population in Thailand, the people we interviewed were treated as secondary after the hill tribe people who hold Thai IDs not only when accessing health care services but also in regard to choosing jobs, accessing schools, and relocating. Stateless people have fewer choices when choosing a job. All professional and high-wage jobs require a Thai ID and school certification. These conditions contribute to the minimized ability of members of the stateless population to obtain a good job. Almost all of the participants reported working as laborers in their hill tribe or on Thai farms for small daily wages.

When individuals live under conditions that cannot fully contribute to obtaining a better life, then individuals become hopeless, or they feel that life is nothing, a feeling that was presented among this group.

A 39-year-old woman stated the following [P#4]:

"I had an appendectomy a couple months ago. With no Thai ID, I was charged 30,000 baht ($1,000). This caused much suffering for me and my husband. If I would have had a Thai ID, then I could have received medical care for free. However, having no Thai ID, I had no better choice."

A 55-year-old man stated the following [P#2]:

"There are many jobs in this country, but I am not qualified to apply for them because I have no Thai ID. I cannot even go outside this district."

A 36-year-old woman stated the following [P#9]:

"When I was young, I truly wanted to go to school like other children. I asked my parents why I could not go and they said it was because we did not have IDs. I felt truly upset, and this affected my life. I lost my opportunity to go to school in my childhood, and now I cannot speak Thai. I cannot get a good job and earn money for my family."

A 15-year-old teenager stated the following [P#1]:

"I have two younger sisters and one younger brother. Although we can go to elementary school, further study at higher levels is impossible. If you do not have a Thai ID, you are not allowed to go to university in Thailand."

Regarding access to school in Thailand, the Thai government encourages all children to attend school for free until the third year of secondary school. However, at all vocational college and university levels, students are asked to show their Thai IDs or passports that identify them as citizens of another country. Members of the stateless population cannot obtain citizenship cards; thus, they do not meet the qualifications to attend higher levels of school.

The lack of Thai IDs directly impacts the lives of members of the stateless population. Having a Thai ID not only permits access to public services but also allows people to attain a good quality of life, which is affected by one's education, income and career.

## III. Stigmatized people perpetuate the stigma

While many hill tribe people have been granted a Thai ID after living in Thailand for some time, other people do not meet the criteria and thus have received no ID. People who hold Thai IDs can work and attend a higher level of school, including the university level. Afterward, many of the hill tribe people who can pursue a higher level of education are then able to work in a good organization in either the public or private sector. Then, some of these individuals perpetuate the stigma to the other tribe members who do not have an ID.

A translator employed in a local hospital located in the hill tribe villages was reported to have engaged in some of the stigma-related behavior faced by the hill tribe people who do not hold Thai IDs while working at the hospital:

"A few weeks ago, I visited a doctor at a hospital. A certain hill tribe man works at this hospital as a translator. He was hired because he has a Thai ID. When I met him, he acted as though we were of a lower status than him. Because they have Thai IDs and are hired in government offices, they act as if they have a higher status than us. I do not like this."

## IV. Stigma coping strategies

The stateless population reported several approaches to handling the stigma problem. Avoiding problems is one of the priority behaviors used when one encounters stigma, as reported by this population. This skill is dominant among members of this population. Examples include not traveling outside the village, not talking with people with whom they are not familiar, and keeping up with any unexpected event that could result in exposure.

A 23-year-old man stated the following [P#8]:

"I am 23 years old, still young and want to enjoy my life, but I have limited time to spend with friends. Even though I want to go to a town to enjoy things like other adolescents, I have to stay in the village. There is a checkpoint to prohibit stateless people from traveling outside the living area, and many police work there to detect illegal migrants. I do not want to have a problem, so I do not go outside the village; I feel sad when I think about this."

A 49-year-old man stated the following [P#22]:

"As I do not have a Thai ID, I am very careful when I meet people. I will not talk to people whom I am not familiar with because I do not know who they are; they might be a spy from the police to catch illegal immigrants. My friend in the village told me that if I am caught, I have to pay a lot of money and stay in jail for many years. Are you a spy?"

A 28-year-old man stated the following [P#26]:

"I am a man who would like to enjoy life and work hard for the money. However, I cannot do anything. I moved from Myanmar 10 years ago by walking through the natural route. My status is still that of a stateless person. I cannot go outside the village the same as other stateless persons because I am scared of being caught by the police. I try not to go to parties or get involved in any risky situations, for example, wedding parties and housewarming celebrations. I stop problems before they happen."

A Thai ID is requested in many situations, especially when participating in government activities. Thus, seeking a way to obtain an ID is one of the most common tactics reported among stateless people. In the field, information was presented that obtaining Thai IDs could be achieved by deceitful approaches.

A 54-year-old woman stated the following [P#20]:

"The government staff surveys the number of stateless population members every year, and I look forward to getting an ID too. I have heard that if you have money and are able to pay for some procedures, you will get an ID easily. However, I am not sure that this information is correct. I, myself, I will not go this way because I do not have much money."

A 28-year-old woman stated the following [P# 36]:

"I heard that people who have an ID had to get close to a member of the government and pay them a lot of money."

She added the following:

"Getting an ID is difficult for me. I do not have enough money to deal with the process of receiving a Thai ID. I also do not know a better way to get it. I and my husband would like to follow the government guidelines about getting a card."

Another tactic for reducing the impact of stigma among the stateless population is practicing the Thai language. Speaking fluent Thai helps to protect against stigma, particularly regarding access to government offices; however, not speaking Thai fluently does not limit one's daily communication with people who speak Thai, such as when individuals sell their farming products.

A 34-year-old woman stated the following [P# 39]:

"When I go to a hospital, I try to use Thai to contact them; otherwise, the staff look down on me. I try to speak Thai, and I try to speak the same as the general Thai accent. This can help me to make myself feel better."

A 47-year-old woman stated the following [P# 17]:

"I know they look down me because I cannot speak Thai. If a person speaks the Thai language fluently, it seems as although we are not the same type of Thai. I can see their behaviors, such as their tone of voice and the look in their eyes. You know? I understand them but I cannot speak to them."

A 37-year-old man stated the following [P# 42]:

"Using the Thai language when selling cabbage and other products from our farm is not difficult. However, every time I go to the district office, I feel nervous and appear to be smaller than I am. I practice speaking in a Thai accent to increase my confidence in talking to them."

Several approaches were reported to address the stigma among the stateless population. Keeping to themselves in villages is a favorite method of avoiding the impact of the stigma. Another approach is to practice speaking Thai. Those who can speak Thai fluently experience little impact from the stigma.

## V. The hope and the brightness

The one and only hope for the stateless population living in Thailand is to have Thai IDs. It is the key to opening everything, including removing the stigma from their lives.

A 58-year-old woman stated the following [P# 3]:

"If I had an ID, it would no longer be difficult to see a doctor in a hospital. No one would ask me about the right to receive treatment."

A 53-year-old woman stated the following [P# 11]:

"When I go to a hospital, I do not dare speak because I cannot speak Thai fluently. I do not understand what a doctor is saying. Moreover, I feel that some staff members look down on me. They always ask me what right I have to access a service. I feel uncomfortable, truly, so getting an ID is the most important thing in my life."

A 27-year-old woman stated the following [P# 40]:

"The most desirable thing in my life is getting a Thai ID. Doing so could help me to get a better job and a better life. This would affect not only my life but also my family's life. Additionally, I would not be part of a secondary population anymore."

Having a Thai ID is a significant factor that affects the likelihood of having a good future for the stateless population living in Thailand. These IDs are used to access medical services and education. Anyone who has a good education will have a better opportunity to get a good job.

## Discussion

The stateless population in Thailand suffer from stigma about which they have various perceptions: having a lower ability to request that their needs be met, not being equal to others, not being able to qualify for health care services and even being ranked below other hill tribe people who do have IDs. "Life is nothing" is reported as the viewpoint of the invisible population in Thailand. Some hill tribe people act as stigma perpetuators among members of the stateless population, which perpetuates the stigma. Keeping within their village, trying to get a Thai ID, and practicing speaking Thai are ways of avoiding the stigma, according to the members of the stateless population interviewed herein. Having a Thai ID was detected as the single goal that would help them overcome the stigma problem.

The stateless population living in Thailand reported several stigma perceptions: having a lower ability to request that their needs be met, not being equal to others, not being able to qualify for health care services and even being ranked below other hill tribe people who do have IDs; in other words, feeling as through life is nothing. These perceptions were presented through the scenarios described by the interviewees. Stateless people in Thailand live as invisible persons. The major factor that forces them to be invisible people is not holding Thai IDs. Thai IDs are used to gain access to all public services in Thailand, including access to health care services [27]. Scenarios of these individuals having no ability to request the help needed were clearly present during the COVID-19 epidemic; i.e., all Thai people who held Thai IDs were financially supported by the central government, while the stateless population remained invisible to policy-makers [28, 29]. All members of the stateless population belong to one of the tribes that live in border villages in Thailand. Those who have not been granted Thai IDs for any reason are defined as the stateless population. Thus, while those who have a Thai ID are helped by the central Thai government and have full rights to access all services, the stateless are not. Some members of the stateless population have been hired to work in either the government or private sector, especially as translators for specific communication requirements. For instance, a hospital hired a stateless population member who spoke tribal languages and facilitated the communication of health professionals; these groups of people are paid approximately 9,000 baht ($300) per month. However, the stateless person who was hired then undertook some derogatory action and perpetuated the stigma among other stateless population members while they were attending the hospital. This perpetuated stigma was attached to the person's peers, which greatly impacted the victims as other stateless population members in the village. Suffering from peer-perpetuated stigma could have multiple impacts on individuals. In contrast, individuals often try not to be part of a group of people who are treated as victims in society. People move away from unpleasant zones of life, which might be a common human behavior.

Several approaches were used to cope with the problem of the stigma faced by people among the stateless population in Thailand. The stateless population tried to remain in private and confidential zones and to avoid interacting with people outside the village. These findings are related to the immigration regulations of the Kingdom of Thailand that prohibit stateless persons from traveling outside of "the restriction areas". This prohibition means they are not allowed to leave the specific areas mentioned. However, under these regulations, if anyone needs to leave the village area, they must have the approval of a government officer. This

process requires a long period of time [30]. The statement of "not talking with people who do not know" reflects the concern of these villages about being caught by the police who work under Thai laws [30, 31]. A study in the United States also reported that illegal immigrants within the country fear being caught by police who work under US laws [32].

Moreover, stateless people try to obtain Thai IDs, which are stamped documents for Thai citizenship. The Constitution of Thailand permits very few ways to obtain an ID [33]; however, doing so is only the way to remove themselves and their subsequent generations from the suffering zone. To preliminarily protect themselves from the stigma, these individuals, especially the younger generation, are improving their skills in regard to speaking Thai. Under Thai regulations, all children living in Thailand are able to fully access primary education until grade 12 [24, 34], regardless of whether they hold Thai IDs [35]. Thus, one of the best options for the new generation is to try to attend school. If they attend school, children will be able to speak Thai fluently, and they will eventually free themselves from the stigma. Therefore, having Thai IDs is the only hope and bright spot among the stateless population in Thailand. If they hold a Thai ID, they can apply for any job position in Thailand, be elected as a politician, and even become the Prime Minster of Thailand.

A few limitations were detected in the study. Some participants could not speak Thai fluently, which might impact the amount and accuracy of the information obtained. However, during the interviews, the researcher asked local people who were fluent in Thai to help collect the information. Two or three participants presented their fears about their status in Thailand to the interviewers. These issues, i.e., real information pertaining to the purpose of the study, were clearly passed to the interviewers through the village leaders. Finally, all of the participants openly provided information to the researchers. In the context of their tremendous stigma-related suffering, some participants were hesitant to talk about their experiences because they did not feel secure doing so. In this situation, the interviewer had to become familiar with these individuals by talking to them for slightly longer than others before being able to collect accurate information.

## Conclusion

The stateless population living in Thailand, especially along the border of Thailand and Myanmar, are treated as an invisible population with limited access to health care services. They have a lower ability to request that their needs be met, they feel unequal to others, they do not qualify for health care services, and they are even ranked below other hill tribe people who have IDs. These are some of the impacts of the stigma that are found in the lives of stateless individuals, including the outlook that "No better options in life", i.e., that one's life is empty. The stateless population are also victimized by other members of the hill tribe who hold Thai IDs and thus serve as reproducers of the stigma.

To address this problem, the Thai government, including all relevant agencies, should promote the obtaining of Thai IDs to these individuals and encourage them to attend school. These individuals would be excellent human resources for the future development of the country. Giving IDs to these people and encouraging them to attend school would also help to ensure that all people in Thailand are granted basic human rights. Another intervention that should be implemented throughout all government offices and services is promoting non-stigma behaviors toward anyone who attempts to receive services or government-office programs. Implementing access to educational institutions without barriers, especially the Thai ID requirement, should be promoted. In addition, creating laws to reduce discrimination to access public services including employment should be implemented.

## Supporting information

**S1 Appendix. Question guideline.**
(DOCX)

**S2 Appendix. Transcript.**
(DOCX)

## Acknowledgments

We thank all the health care workers who worked in the study setting for their assistance in obtaining access and recruiting the participants. We also thank all of the participants for their participation in the study.

## Author Contributions

**Conceptualization:** Siwarak Kitchanapaibul, Tawatchai Apidechkul, Anusorn Udplong.

**Data curation:** Siwarak Kitchanapaibul, Tawatchai Apidechkul, Peeradone Srichan, Thanatchaporn Mulikaburt, Onnalin Singkhorn, Anusorn Udplong, Panupong Upala, Chalitar Chomchoei, Fartima Yeemard, Ratipark Tamornpark, Pilasinee Wongnuch.

**Formal analysis:** Siwarak Kitchanapaibul, Tawatchai Apidechkul, Peeradone Srichan, Thanatchaporn Mulikaburt, Onnalin Singkhorn, Anusorn Udplong, Panupong Upala, Chalitar Chomchoei, Fartima Yeemard, Ratipark Tamornpark, Pilasinee Wongnuch.

**Funding acquisition:** Tawatchai Apidechkul.

**Investigation:** Siwarak Kitchanapaibul, Tawatchai Apidechkul, Peeradone Srichan, Thanatchaporn Mulikaburt, Onnalin Singkhorn, Panupong Upala, Chalitar Chomchoei, Fartima Yeemard, Ratipark Tamornpark, Pilasinee Wongnuch.

**Methodology:** Siwarak Kitchanapaibul, Tawatchai Apidechkul.

**Project administration:** Tawatchai Apidechkul.

**Writing – original draft:** Siwarak Kitchanapaibul, Tawatchai Apidechkul, Peeradone Srichan, Thanatchaporn Mulikaburt, Onnalin Singkhorn, Anusorn Udplong, Panupong Upala, Chalitar Chomchoei, Fartima Yeemard, Ratipark Tamornpark, Pilasinee Wongnuch.

**Writing – review & editing:** Siwarak Kitchanapaibul, Tawatchai Apidechkul, Peeradone Srichan, Thanatchaporn Mulikaburt, Onnalin Singkhorn, Anusorn Udplong, Panupong Upala, Chalitar Chomchoei, Fartima Yeemard, Ratipark Tamornpark, Pilasinee Wongnuch.

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
