## [Decision Letter · Decision Letter 0]

19 Oct 2021

PONE-D-21-27010Status of the stateless population in Thailand: how does stigma matter in their life?PLOS ONE

Dear Dr. Apidechkul,

Thank you for submitting your manuscript to PLOS ONE. After careful consideration, we feel that it has merit but does not fully meet PLOS ONE’s publication criteria as it currently stands. Therefore, we invite you to submit a revised version of the manuscript that addresses the points raised during the review process.

We look forward to receiving your revised manuscript.

Kind regards,

Ghaffar Ali, PhD

Academic Editor

PLOS ONE

Journal Requirements:

3. Thank you for stating the following in the Acknowledgments/ Funding Section of your manuscript: 

This study was supported by a grant from the Center of Excellence for Hill Tribe Health Research, Mae Fa Luang Thailand. The funder had no role in the design of the study, collection, analysis, and interpretation of the data and writing of the manuscript.

Reviewers' comments:

Reviewer's Responses to Questions

**Comments to the Author**

1. Is the manuscript technically sound, and do the data support the conclusions?

Reviewer #1: Yes

Reviewer #2: Partly

2. Has the statistical analysis been performed appropriately and rigorously? 

Reviewer #1: N/A

Reviewer #2: No

3. Have the authors made all data underlying the findings in their manuscript fully available?

Reviewer #1: Yes

Reviewer #2: Yes

4. Is the manuscript presented in an intelligible fashion and written in standard English?

Reviewer #1: Yes

Reviewer #2: Yes

5. Review Comments to the Author

Reviewer #1: The paper is interesting and reflects the significant issue that minorities in Thailand are facing.

The manuscript is well structured.

However, there are some grammatical errors (for example, tenses) and word choices (for example, a mole instead of a spy). The authors need to work more on literature reviews. More discussions on the differences between stigma, prejudice, and discrimination should be added. More reviews on types of stigma: public stigma, self-stigma, & institutional stigma could be a good addition.

The methodology section is well structured and clear.

Some contents in the result section are not matched with headers in the result section, for example, in sections C, D, & E.

The examples related to the Thai language may have a separate section.

It is unclear on the interpretation of “life is nothing.” Is it about meaningless or invisible?

In the conclusion section, the authors may need to suggest more substantial implications or policy recommendations instead of summarizing the findings again.

Reviewer #2: The authors aimed to examine the experience of Thailand's stateless population and how they deal with stigma. The manuscript is a technically sound piece of scientific research that includes data to back up the conclusions. There is a concise statement as well as ethical considerations.

However, there is a lack of theoretical support/discussion. An additional topic of debate might be the concept of stigma in marginalized communities, i.e. Furthermore, the analytical component is demonstrated without a comprehensive literature study or theoretical support/discussion. The analytical focus is solely on Thailand, with only limited implications for other empirical investigations. The author provides no evidence of demographic differences or differing educational backgrounds, which may have an impact on the type/degree of stigma. In many cases, the transcripts were simply inserted without further examination.

Please clearly specify the interviewee selection criteria.

Furthermore, because the definition of a stateless person is "who is not considered as a national by any state under the operation of its law" (The 1954 Convention relating to the Status of Stateless Persons), Simply described, a stateless person is someone who does not have nationality from any country. Thereby, the analytical and conclusion sections, as well as recommendations should be done with caution to avoid recommendations with a tautology-related perspective.

6. PLOS authors have the option to publish the peer review history of their article (what does this mean?). If published, this will include your full peer review and any attached files.

Reviewer #1: No

Reviewer #2: No

---

## [Author Response · Author response to Decision Letter 0]

13 Dec 2021

Response to reviewers’ comments

Dear Plosone,

Thank you very much for given us the opportunity to revise the manuscript. We have carefully checked, cleared, and responded to all points raised by you and reviewers.

Moreover, we have made a bit change on the title from “Status of the stateless population in Thailand: how does stigma matter in their life?” to “Status of the stateless population in Thailand: how does stigma matter in their life ?” which is reported in the cover letter to you as well.

We do hope that you happy in this version, and look forward to hearing from you.

Regards,

TK

PONE-D-21-27010

Status of the stateless population in Thailand: how does stigma matter in their life?

PLOS ONE

Journal Requirements:

: Thank you, we checked and followed the journal guidelines.

: Thank you. This problem is now revised.

3. Thank you for stating the following in the Acknowledgments/ Funding Section of your manuscript:

This study was supported by a grant from the Center of Excellence for Hill Tribe Health Research, Mae Fa Luang Thailand. The funder had no role in the design of the study, collection, analysis, and interpretation of the data and writing of the manuscript.

: Thank you. This information is now revised.

: Please kindly insert the following statement in the funding section:

“Funding

This study was supported by a grant from the Center of Excellence for Hill Tribe Health Research, Mae Fa Luang Thailand (No. 01-2021). The funder had no role in the design of the study; collection, analysis, and interpretation of the data; or writing of the manuscript.”

: Thank you. These files are now revised and updated.

Reviewers' comments:

5. Review Comments to the Author

Reviewer #1:

1) The paper is interesting and reflects the significant issue that minorities in Thailand are facing.The manuscript is well structured. However, there are some grammatical errors (for example, tenses) and word choices (for example, a mole instead of a spy). 

: This manuscript was revised again by American Journal Experts (AJE), reference No 39DA-241D-833B-CF5B-531B 

2) The authors need to work more on literature reviews. More discussions on the differences between stigma, prejudice, and discrimination should be added. More reviews on types of stigma: public stigma, self-stigma, & institutional stigma could be a good addition.

: Thank you for the comment. We added the relevant information on page 3, lines 7-21 with references no 15-19. 

3) The methodology section is well structured and clear.

: Thank you. 

4) Some contents in the result section are not matched with headers in the result section, for example, in sections C, D, & E. The examples related to the Thai language may have a separate section.

: Thank you for this important comment.

: As a team, we carefully discussed this issue and realized that in Parts C and D, two themes were extracted from the analysis: being unable to qualify for health care services and being ranked below other hill tribe people who do hold Thai IDs. We revised all the details of the supported content so it relates to the theme. Meanwhile in Section E, P#9 said she could not get a good job because she did not attend a school when she was young. She also said she could not speak Thai, which was one reason she was not recruited into a good job. We decided to retain the sequence of her story in the manuscript’s contents because we want to ensure that all readers understand the context of the stateless population in Thailand.

: With this important comment, we revised all the contents in the manuscript to ensure that all the information and themes are consistent.

: Thank you very much for the comment and we hope our revisions are clear.

5) It is unclear on the interpretation of “life is nothing.” Is it about meaningless or invisible?

: After carefully reviewing Section E (“Life is nothing”), we completely agree with you that the message lacks interpretation. Thank you very much for the important comment.

: We reviewed all the information and interpreted it. Please see page 11, lines 12-14.

6) In the conclusion section, the authors may need to suggest more substantial implications or policy recommendations instead of summarizing the findings again.

: Thank you for the comment. We improved this section.

Reviewer #2:

1) The authors aimed to examine the experience of Thailand's stateless population and how they deal with stigma. The manuscript is a technically sound piece of scientific research that includes data to back up the conclusions. There is a concise statement as well as ethical considerations.

: Thank you very much.

2) However, there is a lack of theoretical support/discussion. An additional topic of debate might be the concept of stigma in marginalized communities, i.e., Furthermore, the analytical component is demonstrated without a comprehensive literature study or theoretical support/discussion.

: Thank you for the comment.

: Essentially, we reviewed many studies and examined many concentrated conceptual frameworks before undertaking this project. Ultimately, we used the framework integrating normative influence on stigma (FINIS) for the study [15], which turned out to the best fit.

3) The analytical focus is solely on Thailand, with only limited implications for other empirical investigations. The author provides no evidence of demographic differences or differing educational backgrounds, which may have an impact on the type/degree of stigma. In many cases, the transcripts were simply inserted without further examination.

: Thank you for the important comment. This project originated from the stigma experiences among the stateless population in Thailand. During the analysis, we thought carefully about including individual information in the manuscript quotations. However, we realized that inserting increasing amounts of information into the manuscript might violate the ethics of a good human-subject research project. Therefore, we listed the group’s demographics, including education, in Table 1.

: Most of the stateless population in Thailand cannot access education (66.7% had no education and 23.5% attended primary school only), and the stigma impacts may not differ according to their educational level. We hope our explanation is clear.

4) Please clearly specify the interviewee selection criteria.

: The interviewee selection criteria were a) being part of the stateless population, defined as having no Thai ID, and b) experiencing any form of stigma (page 4, lines 19-21).

5) Furthermore, because the definition of a stateless person is "who is not considered as a national by any state under the operation of its law" (The 1954 Convention relating to the Status of Stateless Persons), Simply described, a stateless person is someone who does not have nationality from any country. Thereby, the analytical and conclusion sections, as well as recommendations should be done with caution to avoid recommendations with a tautology-related perspective.

: Thank you so much for such an important comment.

: The first paragraph of the Conclusion was focused on granting Thai IDs because the context of the related laws has changed over time and by case. Therefore, the 1954 Convention relating to the Status of Stateless Persons clearly states the definition of stateless persons, but the new Constitution of Thailand includes much evidence as new criteria to be used for granting IDs. These include making a great contribution to Thai society in any respect, being a person devoted to Thailand through sports, etc. Moreover, the law has been revised to adapt to the new situation of Thai society. Therefore, we included this content in the first paragraph of the Conclusion. We hope that our reasoning is clear. 

TK

Assistant Professor Dr. Tawatchai Apidechkul

Dean, School of Health Science, MFU

Director, Center of Excellence of the Hill Tribe Health Research, WHO-CC

Former Hubert H Humphrey Fellow (2013-2014), Emory University

Global Health Delivery Intensive (Harvard School of Public Health)

---

## [Decision Letter · Decision Letter 1]

27 Jan 2022

PONE-D-21-27010R1Status of the stateless population in Thailand: how does stigma matter in their life?PLOS ONE

Dear Dr. Apidechkul,

Thank you for submitting your manuscript to PLOS ONE. After careful consideration, we feel that it has merit but does not fully meet PLOS ONE’s publication criteria as it currently stands. Therefore, we invite you to submit a revised version of the manuscript that addresses the points raised during the review process.

We look forward to receiving your revised manuscript.

Kind regards,

Ghaffar Ali, PhD

Academic Editor

PLOS ONE

Journal Requirements:

Reviewers' comments:

Reviewer's Responses to Questions

**Comments to the Author**

1. If the authors have adequately addressed your comments raised in a previous round of review and you feel that this manuscript is now acceptable for publication, you may indicate that here to bypass the “Comments to the Author” section, enter your conflict of interest statement in the “Confidential to Editor” section, and submit your "Accept" recommendation.

Reviewer #1: All comments have been addressed

Reviewer #2: All comments have been addressed

2. Is the manuscript technically sound, and do the data support the conclusions?

Reviewer #1: Yes

Reviewer #2: Partly

3. Has the statistical analysis been performed appropriately and rigorously? 

Reviewer #1: N/A

Reviewer #2: Yes

4. Have the authors made all data underlying the findings in their manuscript fully available?

Reviewer #1: Yes

Reviewer #2: Yes

5. Is the manuscript presented in an intelligible fashion and written in standard English?

Reviewer #1: Yes

Reviewer #2: Yes

6. Review Comments to the Author

Reviewer #1: The authors addressed and responded to all of my comments and suggestions. However, I have some comments on their responses.

The name of Section E can be “No better options in life” instead of “Life is nothing.” “No better options in life” may represent the stories in this section clearer. The term “life is nothing” is unclear.

In addition, please be careful on the lettering of section “B. Perceptions of stigma.” Should they all be “d), e)”? Another option is using numbers. It may be easier to structure the sections.

In the conclusion section, the authors suggest that the problems may be alleviated by providing Thai IDs to hill tribe people. They also suggest the promotion of nonstigma behaviors and services toward anyone. In many countries, there are laws on discrimination in healthcare, employment, and education. The authors may explore more on the discrimination laws and regulations.

Like my comment on the subsection “e)”, the term “life is nothing” is unclear. I would suggest that the term “No better options in life” should also be used in the discussion and conclusion.

Reviewer #2: (No Response)

7. PLOS authors have the option to publish the peer review history of their article (what does this mean?). If published, this will include your full peer review and any attached files.

Reviewer #1: No

Reviewer #2: No

---

## [Author Response · Author response to Decision Letter 1]

29 Jan 2022

Response to reviewer comments

Journal Requirements:

: Thank you, we have double check all references. Reference No.7, 9, 16, 17, 18, 26,33, and 34 are revised and completed. 

Reviewers' comments:

Comments to the Author

1. If the authors have adequately addressed your comments raised in a previous round of review and you feel that this manuscript is now acceptable for publication, you may indicate that here to bypass the “Comments to the Author” section, enter your conflict of interest statement in the “Confidential to Editor” section, and submit your "Accept" recommendation.

Reviewer #1: All comments have been addressed

Reviewer #2: All comments have been addressed

2. Is the manuscript technically sound, and do the data support the conclusions?

Reviewer #1: Yes

Reviewer #2: Partly

3. Has the statistical analysis been performed appropriately and rigorously? 

Reviewer #1: N/A

Reviewer #2: Yes

4. Have the authors made all data underlying the findings in their manuscript fully available?

Reviewer #1: Yes

Reviewer #2: Yes

5. Is the manuscript presented in an intelligible fashion and written in standard English?

Reviewer #1: Yes

Reviewer #2: Yes

6. Review Comments to the Author

Reviewer #1: The authors addressed and responded to all of my comments and suggestions. However, I have some comments on their responses.

The name of Section E can be “No better options in life” instead of “Life is nothing.” “No better options in life” may represent the stories in this section clearer. The term “life is nothing” is unclear.

: Thank you, we agreed with you and have changed accordingly, please see page 11, line 4.

In addition, please be careful on the lettering of section “B. Perceptions of stigma.” Should they all be “d), e)”? Another option is using numbers. It may be easier to structure the sections.

: To make sure that the sequence is not difficult for readers, we have revised by using Greek alphabet (I, II...) to the main theme and use English in sub-theme. Thank you

In the conclusion section, the authors suggest that the problems may be alleviated by providing Thai IDs to hill tribe people. They also suggest the promotion of nonstigma behaviors and services toward anyone. In many countries, there are laws on discrimination in healthcare, employment, and education. The authors may explore more on the discrimination laws and regulations.

: Thank you for great comment here, we have added information on creating and implementing relevant laws on discrimination protection in access healthcare services in this section, please see in page 18, lines 27-29.

Like my comment on the subsection “e)”, the term “life is nothing” is unclear. I would suggest that the term “No better options in life” should also be used in the discussion and conclusion.

: We agreed with you and revised.

Reviewer #2: (No Response)

7. PLOS authors have the option to publish the peer review history of their article (what does this mean?). If published, this will include your full peer review and any attached files.

Do you want your identity to be public for this peer review? For information about this choice, including consent withdrawal, please see our Privacy Policy.

Reviewer #1: No

Reviewer #2: No

Thank you,

TK

---

## [Decision Letter · Decision Letter 2]

21 Feb 2022

Status of the stateless population in Thailand: how does stigma matter in their life?

PONE-D-21-27010R2

Dear Dr. Apidechkul,

We’re pleased to inform you that your manuscript has been judged scientifically suitable for publication and will be formally accepted for publication once it meets all outstanding technical requirements.

Kind regards,

Ghaffar Ali, PhD

Academic Editor

PLOS ONE

Additional Editor Comments (optional):

Reviewers' comments:

Reviewer's Responses to Questions

**Comments to the Author**

1. If the authors have adequately addressed your comments raised in a previous round of review and you feel that this manuscript is now acceptable for publication, you may indicate that here to bypass the “Comments to the Author” section, enter your conflict of interest statement in the “Confidential to Editor” section, and submit your "Accept" recommendation.

Reviewer #1: All comments have been addressed

2. Is the manuscript technically sound, and do the data support the conclusions?

Reviewer #1: Yes

3. Has the statistical analysis been performed appropriately and rigorously? 

Reviewer #1: N/A

4. Have the authors made all data underlying the findings in their manuscript fully available?

Reviewer #1: Yes

5. Is the manuscript presented in an intelligible fashion and written in standard English?

Reviewer #1: Yes

6. Review Comments to the Author

Reviewer #1: The authors already responded all my comments and suggestions. However, there is a issue on formatting under section II. The authors should make their decision on the style of subsection. Should they be a), b), c), d), e) or a., b., c., d., e.?

7. PLOS authors have the option to publish the peer review history of their article (what does this mean?). If published, this will include your full peer review and any attached files.

Reviewer #1: No

---

## [Editor Report · Acceptance letter]

4 Mar 2022

PONE-D-21-27010R2 

Status of the stateless population in Thailand: how does stigma matter in their life? 

Dear Dr. Apidechkul:

I'm pleased to inform you that your manuscript has been deemed suitable for publication in PLOS ONE. Congratulations! Your manuscript is now with our production department. 

Kind regards, 

on behalf of

Prof. Ghaffar Ali 

Academic Editor

PLOS ONE